# The glutathione import system satisfies the *Staphylococcus aureus* nutrient sulfur requirement and promotes interspecies competition

Joshua M. Lensmire[1], Michael R. Wischer[1], Cristina Kraemer-Zimpel[1], Paige J. Kies[1], Lo Sosinski[2], Elliot Ensink[3], Jack P. Dodson[1], John C. Shook[1], Phillip C. Delekta[1], Christopher C. Cooper[4], Daniel H. Havlichek, Jr.[4], Martha H. Mulks[1], Sophia Y. Lunt[3,5], Janani Ravi[6], Neal D. Hammer[1]*

**1** Department of Microbiology and Molecular Genetics, Michigan State University, East Lansing, Michigan, United States of America, **2** Department of Pathobiology and Diagnostic Investigation, Michigan State University, East Lansing, Michigan, United States of America, **3** Department of Biochemistry and Molecular Biology, Michigan State University, East Lansing, Michigan, United States of America, **4** Department of Medicine, Division of Infectious Disease, Michigan State University, East Lansing, Michigan, United States of America, **5** Department of Chemical Engineering and Materials Science, Michigan State University, East Lansing, Michigan, United States of America, **6** Department of Biomedical Informatics, Center for Health Artificial Intelligence, University of Colorado Anschutz, Aurora, Colorado, United States of America

* hammern2@msu.edu

**Data Availability Statement:** Raw data supporting the figures can be accessed in S1 Data Table.

## Abstract

Sulfur is an indispensable element for bacterial proliferation. Prior studies demonstrated that the human pathogen *Staphylococcus aureus* utilizes glutathione (GSH) as a source of nutrient sulfur; however, mechanisms of GSH acquisition are not defined. Here, we identify a five-gene locus comprising a putative ABC-transporter and predicted γ–glutamyl transpeptidase (*ggt*) that promotes *S. aureus* proliferation in medium supplemented with either reduced or oxidized GSH (GSSG) as the sole source of nutrient sulfur. Based on these phenotypes, we name this transporter operon the **g**lutathione **i**mport **s**ystem (*gisABCD*). Ggt is encoded within the *gisBCD* operon, and we show that the enzyme is capable of liberating glutamate using either GSH or GSSG as substrates, demonstrating it is a *bona fide* γ–glutamyl transpeptidase. We also determine that Ggt is expressed in the cytoplasm, representing only the second example of cytoplasmic Ggt localization, the other being *Neisseria meningitidis*. Bioinformatic analyses revealed that *Staphylococcus* species closely related to *S. aureus* encode GisABCD-Ggt homologs. However, homologous systems were not detected in *Staphylococcus epidermidis*. Consequently, we establish that GisABCD-Ggt provides a competitive advantage for *S. aureus* over *S. epidermidis* in a GSH- and GSSG-dependent manner. Overall, this study describes the discovery of a nutrient sulfur acquisition system in *S. aureus* that targets GSSG in addition to GSH and promotes competition against other staphylococci commonly associated with the human microbiota.

**Funding:** This work was supported by the National Institute of Allergy and Infectious Diseases of the National Institutes of Health, R01 AI139074 and R21 AI142517, and the American Heart Association (16SDG30170026) to NDH. SYL was supported by funding from the National Cancer Institute of the National Institutes of Health (R01CA270136) and the National Institute of Environmental Health Sciences of the National Institutes of Health (R01ES030695), as well as the National Science Foundation (CAREER Grant No. CBET 1845006). The funders had no role in study design, data collection and analysis, decision to publish, or preparation of the manuscript.

**Competing interests:** The authors have declared that no competing interests exist.

## Author summary

Glutathione (GSH) is an abundant sulfur-containing antioxidant present in mammalian tissues. As such, it is a potential source of nutrient sulfur for human pathogens like *Staphylococcus aureus*. However, mechanisms of *S. aureus* GSH acquisition are undefined. Herein, we describe a transport system that supports proliferation of *S. aureus* in medium supplemented with reduced or oxidized (GSSG) GSH and we name it the **G**SH **i**mport **s**ystem or GisABCD. We determine that GSH and GSSG can be catabolized by Ggt and demonstrate that the enzyme resides within the staphylococcal cytoplasm, indicating that GSH and GSSG are imported intact. Lastly, we reveal that GisABCD-Ggt facilitates competition over *Staphylococcus epidermidis* in a GSH- and GSSG-dependent manner, demonstrating GisABCD-Ggt promotes nutrient sulfur acquisition between commensal bacteria.

## Introduction

To colonize the host, bacterial pathogens and commensals are limited to metabolites present in tissues to fulfill nutritional requirements. While strategies employed by bacteria to acquire nutrient transition metals such as iron have been an intense area of research, studies defining mechanisms of nutrient sulfur procurement from host environments are garnering heightened interest [1–6]. Sulfur is essential due to its capacity to fluctuate between redox states and therefore catalyze numerous cellular reactions [7,8]. Ultimately, cells require sulfur to synthesize cysteine (Cys) as it is the fulcrum of sulfur metabolism by serving as an intermediate for methionine (Met) and sulfur-containing cofactors such as Fe-S clusters [8–11]. In host cells and some bacterial species, Cys is also required to generate the low molecular weight thiol glutathione (GSH) [12].

GSH concentrations range between 0.5 and 10 mM in mammalian tissues, making it a relatively abundant source of nutrient sulfur for invading pathogens [3,12,13]. In addition to Cys, GSH consists of glutamate and glycine. A unique γ-peptide bond links the glutamate γ-carboxyl to the Cys amine. To maintain Cys reservoirs, organisms rely on GSH catabolism via the γ-glutamyl cycle [3,12]. Liberation of Cys from GSH is a two-step process that requires γ-glutamyl transpeptidase (Ggt), a specialized protease conserved in all domains of life due to its role in the γ-glutamyl cycle [12,14,15]. Ggt is localized within the Gram-negative periplasm or the periphery of eukaryotic cells where it degrades endogenous GSH [16–18], but it has also been shown to fulfill the nutritional sulfur requirement of *Francisella tularensis* [3,19].

*Staphylococcus aureus* is the leading cause of superficial and invasive bacterial diseases in the United States and Europe [20,21]. Strategies *S. aureus* employs to obtain nutrient sulfur during pathogenesis are largely unknown. Previous work demonstrated that reduced cysteine (Cys), oxidized cysteine or cystine (CSSC), sodium sulfide, thiosulfate, or GSH stimulated *in vitro* proliferation of *S. aureus* [22]. Further investigation revealed that the TcyP and TcyABC transporters support Cys and CSSC utilization; however, mechanisms of GSH acquisition have not been defined [22,23]. *S. aureus* does not synthesize GSH but encodes a putative Ggt. This fact supports the hypothesis that staphylococcal Ggt catabolizes exogenous host GSH, liberating Cys as a means to satisfy the nutrient sulfur requirement [24]. Here we demonstrate that *S. aureus* utilizes oxidized GSH (GSSG) as a nutrient sulfur source and isolate mutants that display significantly decreased proliferation in a medium supplemented with GSSG or GSH as the sole source of nutrient sulfur. These mutants harbor transposon (Tn) insertions within a five-gene locus, *SAUSA300_0200–0204*, that encodes a predicted ATP-binding-cassette (ABC)

transporter (*SAUSA300_0200–0203*) and putative Ggt (*SAUSA300_0204*). Based on the mutant proliferation defects observed in GSH- or GSSG-supplemented media, we name this transporter the **G**lutathione **i**mport **s**ystem (GisABCD). We determine that *S. aureus* Ggt is localized within the cytoplasm and that the recombinant enzyme cleaves both GSH and GSSG. A search for GisABCD-Ggt across Firmicutes revealed that only a select clade within the *Staphylococcus* genus, one that excludes *S. epidermidis*, encodes homologues of the system. Consistent with this finding, *S. aureus* outcompetes *S. epidermidis* in GSSG- or GSH-supplemented conditions in a GisABCD-Ggt-dependent manner. Therefore, this newly described nutrient sulfur acquisition system provides a competitive advantage for *S. aureus* over other staphylococci associated with the human microbiota.

## Results

### *S. aureus* proliferates in medium supplemented with GSSG as the sole source of nutrient sulfur

A previous study qualitatively reported that *S. aureus* proliferates on a chemically defined agar medium supplemented with GSH as the sole sulfur source, indicating that the abundant host metabolite is a viable source of nutrient sulfur [22]. However, *S. aureus* likely encounters both reduced and oxidized GSH (GSSG) as the pathogen induces a potent oxidative burst during infection [25]. Therefore, we hypothesized that *S. aureus* also utilizes GSSG as a source of nutrient sulfur. To quantitatively assess GSH and GSSG utilization as sulfur sources by *S. aureus*, a chemically defined medium, referred to as PN, was employed [26]. PN contains sulfate ($MgSO_4$) and methionine (Met), but *S. aureus* is not capable of assimilating sulfate or utilizing Met to generate Cys due to an apparent lack of a methionine *S*-methyltransferase homologue [9]; thus, oxidized cysteine or cystine (CSSC) is typically added as the source of nutrient sulfur [22,27]. In keeping with this, a USA300 LAC strain of *S. aureus* (JE2) exhibits substantially decreased proliferation in PN that lacks CSSC (**Fig 1A**). Notably, replacing CSSC with either 50 μM GSH or 25 μM GSSG stimulates robust *S. aureus* proliferation (**Fig 1A**). To determine whether utilization of GSSG is conserved throughout the species, we examined proliferation of clinical isolates in GSSG-supplemented medium. Growth of methicillin-

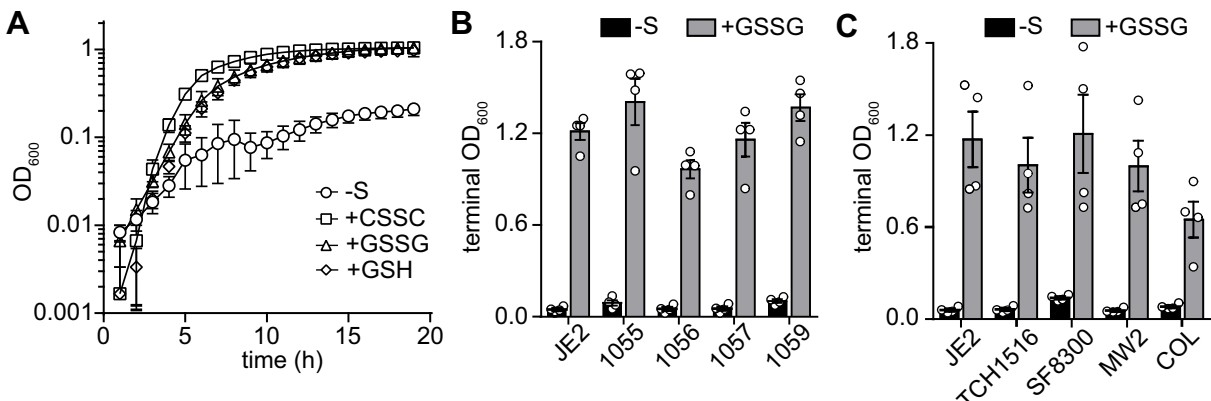

**Fig 1. Supplementation of GSSG as the sole source of nutrient sulfur supports proliferation of *S. aureus*. A.** *S. aureus* JE2 was cultured in PN medium lacking a viable sulfur source (-S) or supplemented with CSSC (25 μM), GSSG (25 μM), or GSH (50 μM). **B.** JE2 and either MSSA or MRSA clinical isolates were grown in PN supplemented without a viable sulfur source (-S) or 25 μM GSSG for 19 hrs. **C.** *S. aureus* strains were cultured in PN medium supplemented without sulfur (-S) or 25 μM GSSG and cultured for 25 h. Bars depict the mean terminal optical density at 600 nm ($OD_{600}$), and circles represent individual replicate terminal $OD_{600}$. The mean of at least three independent trials and error bars representing ± 1 standard error of the mean are presented.

susceptible and methicillin-resistant clinical isolates was quantified in PN supplemented with GSSG as the sole sulfur source. Compared to PN lacking a viable sulfur source, GSSG supplementation stimulates proliferation (**Figs 1B and S1A**). GSSG supplementation also promotes growth of other *S. aureus* strains (**Figs 1C and S1B**). *S. aureus* utilization of GSSG expands the number of sulfur-containing metabolites present in host tissues that are capable of supporting its proliferation.

## The SAUSA300_0200–0204 locus supports *S. aureus* utilization of GSH and GSSG as sulfur sources

To determine genetic factors required for *S. aureus* utilization of GSSG as a sulfur source, we screened the Nebraska Transposon Mutant Library for mutants that exhibit decreased proliferation in PN medium supplemented with 25 µM GSSG as the sole source of sulfur [28]. Five GSSG proliferation-impaired mutants were identified in the screen, each harboring an independent Tn insertion in one of five genes present in the *SAUSA300_0200-ggt* locus (**Fig 2A**). The *ggt* gene (*SAUSA300_0204*) encodes an annotated γ-glutamyl transpeptidase, an enzyme that catabolizes GSH in other organisms [12,14,15]. Notably, *SAUSA300_0200–0203* encodes a putative nickel-peptide ABC transporter. Backcrossing these Tn-inactivated genes into an otherwise wild type (WT), JE2 strain significantly decreased proliferation in medium supplemented with 25 µM GSSG (**Fig 2B**). The Tn mutants also displayed decreased proliferation in PN supplemented with 50 µM GSH as the sole sulfur source (**Fig 2C**). However, the mutant strains demonstrated WT-like growth in medium supplemented with 25 µM CSSC or in a rich medium, indicating the proliferation defect is specific to GSH and GSSG (**Fig 2C** inset). To address complications associated with auto-oxidation of GSH to GSSG in aerobic environments, we tested anaerobic proliferation in media supplemented with GSH or GSSG. Mutant strains harboring a Tn in *SAUSA300_0201* or an in-frame deletion of all five genes were used

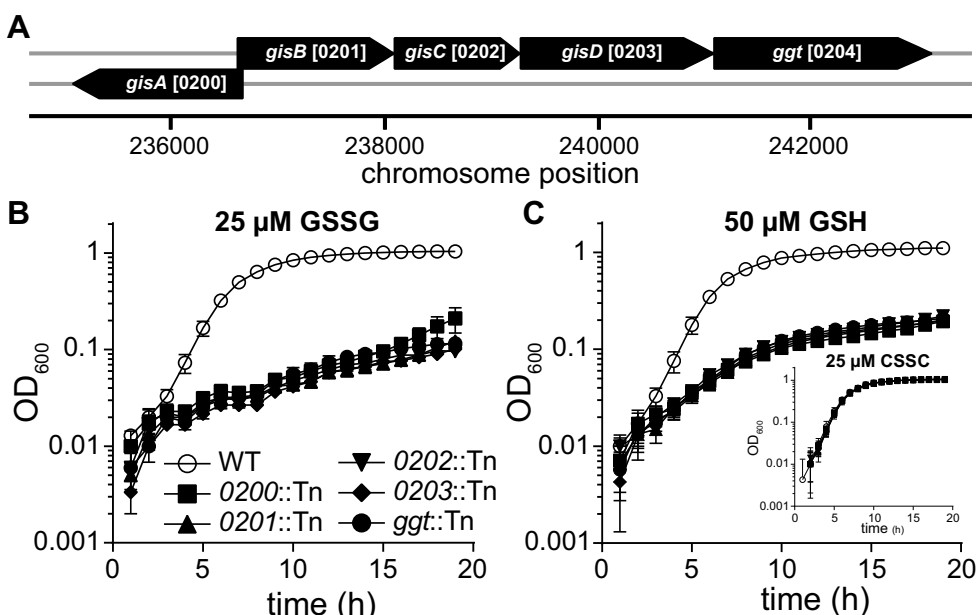

**Fig 2. SAUSA300_0200-*ggt* supports *S. aureus* utilization of GSSG and GSH as sources of nutrient sulfur. A.** Orientation of SAUSA300_0200-*ggt* encoded within the *S. aureus* genome. **B** and **C.** Strains cultured in PN supplemented with 25 µM GSSG (B), 50 µM GSH (C), or 25 µM CSSC (C-inset). The mean of at least three independent trials and error bars representing ± 1 standard error of the mean are presented.

in this assessment. In these conditions, the mutant strains proliferate to WT levels in PN supplemented with CSSC but display diminished growth upon GSH or GSSG supplementation (**S2 Fig**). This finding confirms that *SAUSA300_0200–ggt* supports *S. aureus* utilization of both GSH and GSSG as distinct sources of nutrient sulfur. Complementation experiments tested whether proliferation of a *ggt* mutant cultured in PN medium supplemented with GSH or GSSG could be restored by providing WT or a C-terminal His-tagged *ggt* encoded on a plasmid. *ggt* mutant strains harboring either plasmid display WT-like growth, confirming that decreased proliferation in GSSG- or GSH-supplemented medium is due to genetic inactivation of *ggt* (**S3 Fig**).

Given the decreased proliferation exhibited by the mutants in GSH- or GSSG-supplemented PN media and the fact that *SAUSA300_0200–SAUSA300_0204* encodes a putative ABC-transporter with a predicted γ-glutamyl transpeptidase (Ggt, SAUSA300_0204), we propose to rename *SAUSA300_0200–SAUSA300_0203* the **g**lutathione **i**mport **s**ystem (*gisABCD*) (**Fig 2A**). Domain architecture analysis of the GisABCD-Ggt system reveals that GisA contains ATP-binding cassette domains (**S4 Fig**). Consistent with the prediction, recombinant GisA purified from *Escherichia coli* exhibits ATP hydrolysis activity (**S5 Fig**). GisB and GisC contain nine predicted transmembrane regions, consistent with their membrane-bound annotation, while GisD contains a signal peptide with a putative lipid attachment site (**S4 Fig**). Finally, our domain architecture analysis predicts that *ggt* encodes a γ-glutamyl transpeptidase domain spanning most of the protein (**S4 Fig**).

We next sought to determine whether GisABCD promotes *S. aureus* GSSG acquisition. To test this, liquid chromatography-tandem mass spectrometry (LC-MS/MS) was used to quantify GSSG in WT, a mutant strain harboring an inframe deletion of *gisABCD-ggt* (Δ*gis*), and the previously described *ggt* transposon mutant. Cells were cultured to mid-exponential phase using 25 μM CSSC as the sulfur source, washed, and then incubated in the absence or presence of 25 μM GSSG for 5 min. In keeping with the fact that *S. aureus* does not synthesize GSH and therefore is incapable of using it as its low molecular weight thiol, we were unable to detect GSSG in the absence of supplementation across the three strains (**Fig 3**) [29]. Importantly, GSSG levels in supplemented WT were significantly increased compared to the supplemented Δ*gis* mutant (**Fig 3**). There was no significant difference between GSSG levels in the transporter encoding *ggt* mutant compared to WT upon exposure (**Fig 3**). Taken together with the proliferation phenotypes of the *gis* mutants and the bioinformatic evidence, these data support the conclusion that GisABCD functions as an importer.

## Ggt hydrolyzes GSH and GSSG γ-peptide bonds and localizes to the staphylococcal cytoplasm

A hallmark of γ-glutamyl transpeptidases is their capacity to cleave the GSH γ–peptide bond, liberating glutamate. To validate the domain prediction, we sought to determine whether *S. aureus* Ggt cleaves the γ–peptide bond linking glutamate to Cys in GSH and GSSG [30]. C-terminal His-tagged recombinant Ggt (rGgt) was expressed and purified from *E. coli* (**S6 Fig**). In other species, Ggt is translated as an inactive polypeptide that is auto-catalytically cleaved to generate approximate 40 kDa and 35 kDa subunits [30,31]. In keeping with this, a tripartite banding pattern consisting of full-length pro-Ggt (75 kDa) and smaller, mature enzyme subunits (40 kDa and 35 kDa) are observed (**S6 Fig**). Mature Ggt cleaves GSH by attacking the glutamyl residue, transferring it to the enzyme. Ultimately, water hydrolyzes the γ-peptide bond, liberating glutamate [32]. Therefore, to quantify *S. aureus* Ggt γ-glutamyl transpeptidase activity, rGgt was incubated with increasing concentrations of GSH or GSSG and glutamate release was measured via mass spectrometry. Glutamate was detected in reactions containing

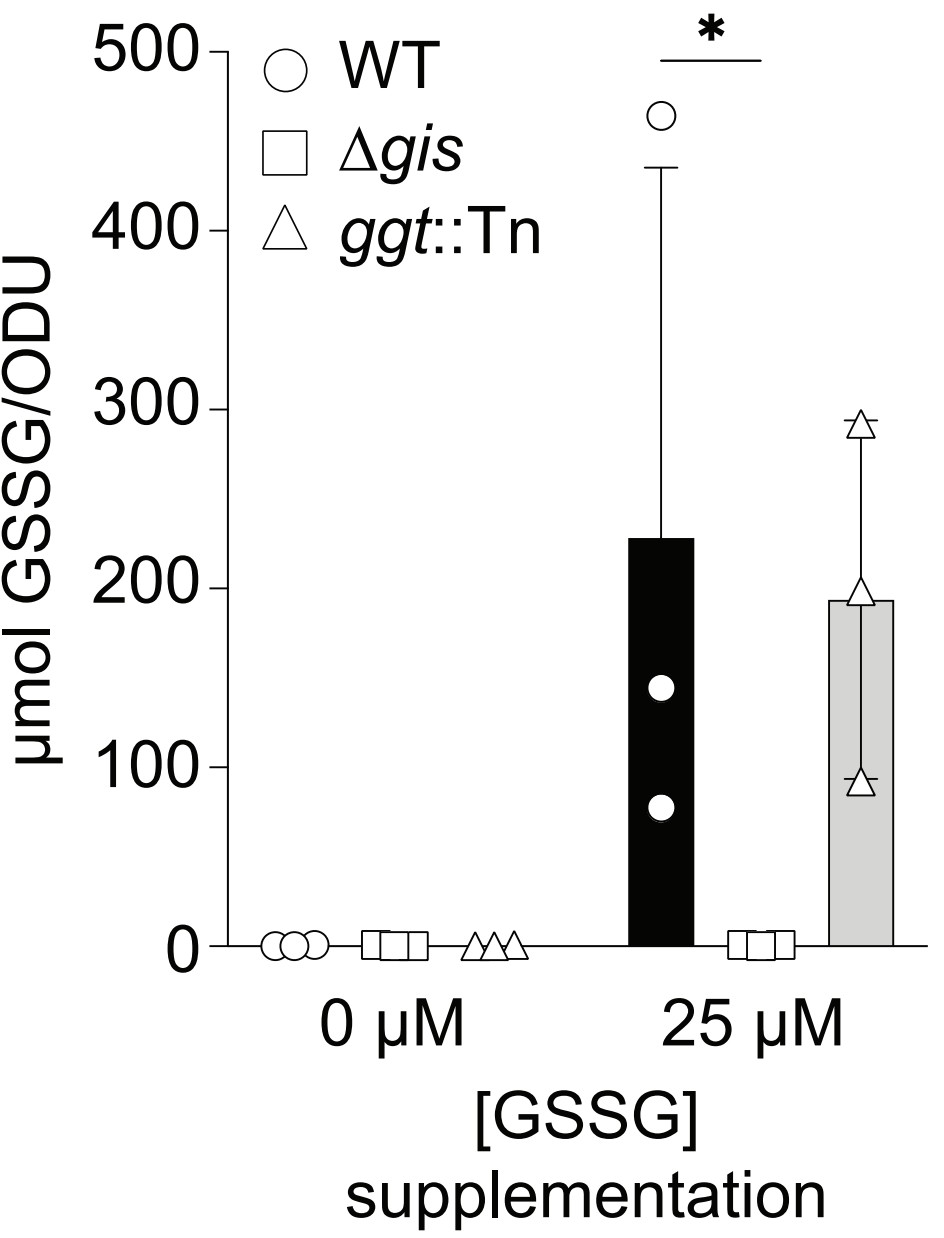

**Fig 3. GisABCD promotes acquisition of GSSG.** Levels of GSSG in WT, Δ*gis*, or *ggt* mutant cells after a 5 min exposure measured by liquid chromatography-tandem mass spectrometry (LC-MS/MS). Samples were normalized by $OD_{600}$ (ODU). The mean and standard deviation of three biological replicates are presented. * *P*-value < 0.05 as determined by two-way ANOVA with Dunnett's multiple comparisons test.

rGgt incubated in the presence of either GSSG or GSH (**Fig 4A and 4B**). Importantly, glutamate was not detected in reactions lacking substrate or rGgt, indicating glutamate release resulted from enzymatic activity. $K_m$ values of Ggt for GSSG and GSH were determined to be 38.6 μM and 58.5 μM, respectively. These values are similar to previously reported Ggt homologues expressed in other organisms (e.g. the $K_m$ of *E. coli* Ggt for GSH is 29 μM [33]). *S. aureus* rGgt $V_{max}$ for GSSG and GSH are 1.1 μmoles min$^{-1}$ and 1.0 μmoles min$^{-1}$, respectively. These data support *in silico* predictions and provide a molecular explanation for the *ggt* mutant

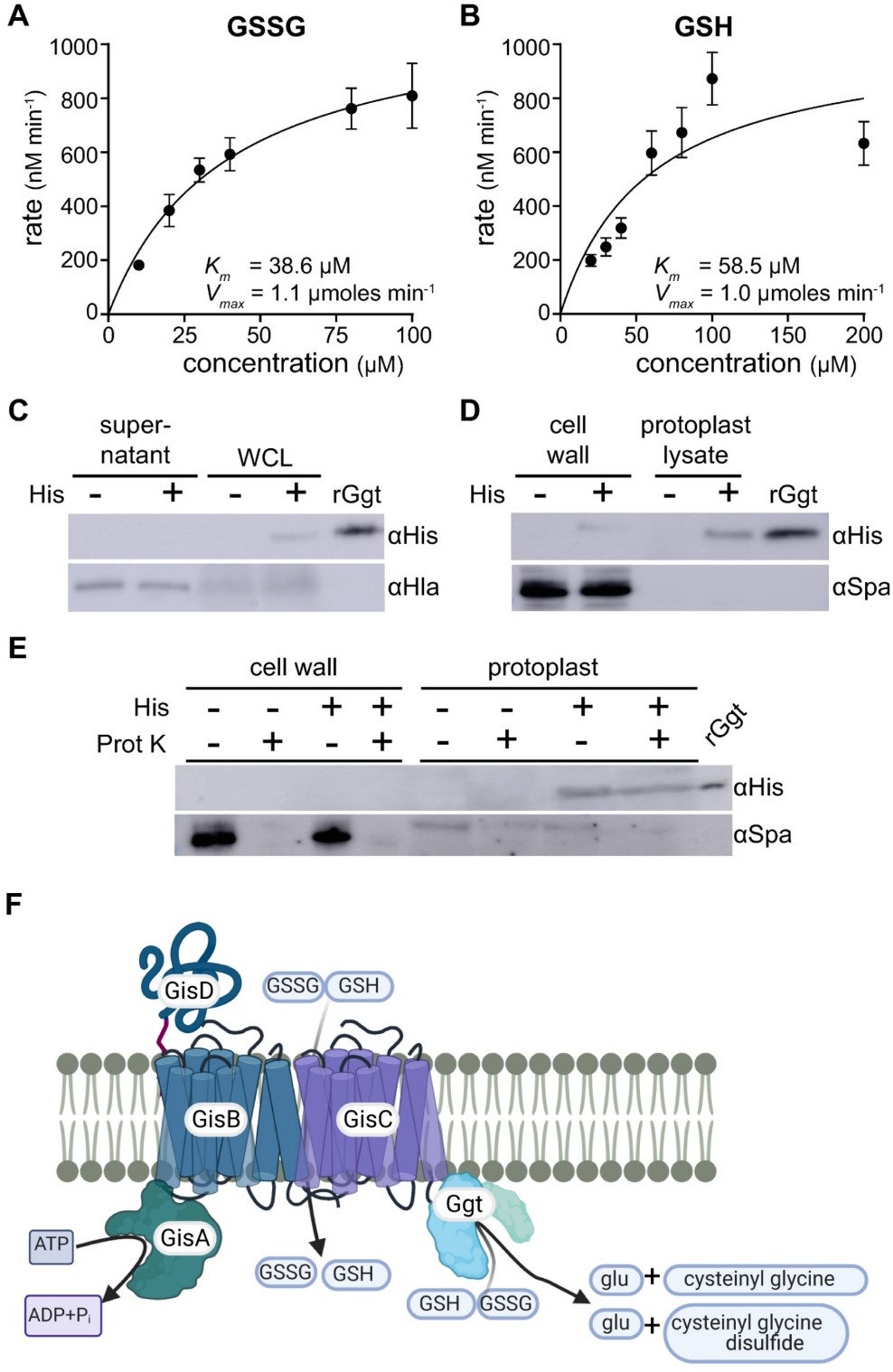

**Fig 4. *S. aureus* Ggt liberates glutamate from GSH and GSSG in the cytoplasm. A** and **B.** rGgt was incubated with indicated concentrations of GSSG (A) or GSH (B). Mean glutamate release per minute was measured using four independent rGgt protein preparations. Glutamate released per minute was calculated and data were fit with the Michaelis Menten equation using GraphPad Prism. Error bars represent ± standard error of the mean. **C.** Supernatant or whole cell lysate (WCL) fractions of mid-exponential *S. aureus ggt*::Tn harboring pOS1 P$_{lgt}$::*ggt* (His -) or pOS1 P$_{lgt}$::

*ggt*-His (His +) probed with an anti-His tag antibody (αHis) or anti-Hla antibody (αHla). His-tagged recombinant Ggt (rGgt) was used as a positive control. **D.** Cell wall and protoplast lysate fractions generated from strains cultured to mid-exponential phase expressing His-tagged (+) or untagged (-) *ggt* were probed with αHis antibodies or anti-protein A antibodies (αSpa). **E.** Cell wall or intact protoplasts derived from the indicated cells were incubated in the presence (Prot K+) or absence (Prot K -) of proteinase K (Prot K). To monitor fractionation and Prot K activity, samples were probed with αSpa. **F.** Bioinformatic predictions and experimental evidence support the presented model of *S. aureus* import and catabolism of exogenous GSH and GSSG. The predicted substrate-binding protein, GisD, binds GSH or GSSG in the extracellular milieu, which are transported into the cytoplasm by the transmembrane permease complex, GisBC. GisA hydrolysis of ATP provides energy needed for import. Finally, GSH and GSSG are cleaved in the cytoplasm by Ggt, generating glutamate and cysteinyl-glycine or cysteinyl-glycine disulfide, depending on the substrate. The model illustration was created using BioRender.

proliferation defect in medium supplemented with GSH or GSSG as sources of nutrient sulfur —*ggt* mutant cells fail to initiate Cys liberation due to an inability to hydrolyze the GSH or GSSG γ-peptide bond. Next, we sought to determine whether hydrolysis of GSH and GSSG occurs intracellularly or extracellularly by probing Ggt localization in *S. aureus* subcellular fractions.

*Bacillus* spp. secrete Ggt [19]; however, structural and cellular localization predictions of *S. aureus* Ggt did not detect canonical secretion signal sequences within the primary sequence (**S4 Fig**). For example, SignalP predictions detected considerably low likelihoods of signal peptide, twin-arginine translocation (TAT) signal peptide, or lipoprotein signal peptide sequences (0.007, 0.003, 0.008, respectively). TatP 1.0 also failed to predict a signal peptide [34,35]. Ggt localization varies across organisms, but in this case, localization has implications for the substrate of GisABCD. Extracellular Ggt supports a model whereby GisABCD imports Ggt products, whereas intracellular Ggt suggests GisABCD imports GSH and GSSG intact. Evidence for the latter is supported by the presence of GSSG in GSSG-exposed WT but not Δ*gis* mutant cells (**Fig 3**). We used the previously described His-tagged Ggt expression vector (Ggt-His) that functionally complements the *ggt*::Tn mutant (**S3 Fig**) to determine subcellular localization of the enzyme. *S. aureus* cells expressing native or Ggt-His were cultured in PN supplemented with 25 μM GSSG, collected at mid-exponential phase, and fractionated into supernatant and whole cell lysate (WCL). An αHis tag antibody (αHis) was used to monitor Ggt-His within each fraction. rGgt served as a size comparison control. Bands corresponding to Ggt-His were not detected in the supernatant fractions; however, a band at ~35kDa was observed in both the Ggt-His WCL and rGgt samples. This band is specific to Ggt-His as it was not observed in WCL generated from cells expressing Ggt lacking the His-tag and corresponds to the rGgt subunit containing the His-tag. Presence of Ggt-His signal within the WCL fraction supports the conclusion that Ggt is cell-associated (**Fig 4C**). To further resolve Ggt localization, cells were fractionated into peptidoglycan cell wall and protoplast fractions. Protoplasts were lysed, generating protoplast lysate that contained the cytoplasm and membrane. A His-dependent ~35 kDa signal was increasingly apparent in the protoplast lysate fraction compared to the cell wall fraction when equivalent levels of total protein are assessed (**Fig 4D**). Protein A (Spa), a protein covalently linked to the peptidoglycan cell wall, was used as a fractionation control. To rule out peripheral, external association of Ggt with the outer leaflet of the plasma membrane, intact protoplasts were isolated and sensitivity to proteinase K (Prot K) was monitored. Spa was used to control for fractionation and Prot K activity. As expected, Spa predominantly localizes to the cell wall and was sensitive to Prot K (**Fig 4E**). On the other hand, a His-dependent band corresponding to Ggt was present in the protoplast fraction in samples treated with or without Prot K (**Fig 4E**). This finding indicates that the membrane shields Ggt from Prot K, supporting the conclusion that the protein resides within the cytoplasm. Taken together, the lack of a secretion signal sequence, cell lysate association, and Prot

K protection support a model whereby GSH and GSSG are imported intact and catabolized in cytoplasm, liberating Cys to satisfy the nutrient sulfur requirement (**Fig 4F**). Cytoplasmic localization of Ggt has been reported in only one other bacterial pathogen, *Neisseria meningitidis* [18].

## GisABCD-Ggt is not required for systemic infection of *S. aureus*

Given its critical role in nutrient sulfur acquisition *in vitro* and the abundance of GSH in host tissues, we next tested the potential role of GisABCD-Ggt in *S. aureus* pathogenesis using a systemic murine model of infection. Contrary to our expectations, we found that mice infected with a *gisB*::Tn mutant contained equivalent bacterial burdens compared to WT-infected animals (**S7 Fig**). This result indicates that GisABCD-Ggt is dispensable for *S. aureus* pathogenesis during systemic infection. There are at least two possibilities for the WT-like virulence exhibited by the *gisB*::Tn mutant. First, acquisition of other sulfur sources, such as Cys or CSSC, is sufficient to fulfill nutrient sulfur acquisition in the absence of GSH or GSSG scavenging. In keeping with this, a previous study showed that the TcyP and TcyABC cysteine transporters support *S. aureus* fitness during heart and liver colonization [4]. The second possible explanation is that *S. aureus* encodes multiple GSH and GSSG transporters. In fact, while GisABCD-Ggt supports proliferation of *S. aureus* in micromolar concentrations of GSSG or GSH, increasing GSH concentrations to levels present in host tissues restores Δ*gis* mutant proliferation in PN medium (**S8 Fig**). However, amplifying GSSG concentrations did not stimulate Δ*gis* mutant proliferation. These results support the conclusion that GisABCD-Ggt is an absolute requirement for GSSG utilization but that another, potentially low-affinity, GSH transporter is also active in this pathogen.

## GisABCD-Ggt is conserved in select Firmicutes

To define a function for GisABCD-Ggt beyond systemic host colonization, we traced the conservation and evolution of the system throughout Firmicutes using a molecular evolution and phylogenetic approach [36]. Due to the ubiquity of ABC-transporters across bacterial genera, we first focused on potential Ggt homologues. We found that many genera encode Ggt homologues, including *Bacillus*, *Gracilibacillus*, *Lysinibacillus*, and *Brevibacterium*; however, subsequent identification of potential GisABCD homologues was limited to a small subset (**S9 Fig**). Overall distribution of GisABCD-Ggt homologues across Firmicutes revealed six distinct clusters. Cluster 3 is the least populated, containing species encoding only Ggt homologues (e.g., *Clostridium tetanomorphum*). Firmicutes in clusters 2 and 4 encode Ggt and either a GisB or a GisC homologue, respectively (e.g., *Gracibacillus thailandensis*). Genomes in Cluster 5 contain GisB, GisC, and Ggt homologues (e.g., *Bacillus licheniformis*; **S9 Fig**). Cluster 6 includes a few bacilli species that encode nearly the complete *S. aureus* GisABCD-Ggt system (e.g., *B. subtilis*), but Cluster 1 stands apart as these genomes encode the full GisABCD-Ggt system similar to the *S. aureus* query sequences (>80% similarity; e.g., *Staphylococcus simiae*). Notably, this cluster is restricted to members of the *Staphylococcus* genus (**S9 Fig**).

Homologues of a complete GisABCD-Ggt system were exclusively identified in the *S. aureus*-related cluster complex (Cluster 6)—which includes *S. argenteus*, *S. schweitzeri*, and *S. simiae* (**Fig 5A**) [37,38]. Conservation rapidly diverged with increasing 16S rRNA phylogenetic distance as the next closest related species, *S. epidermidis*, lacks apparent GisB and GisD homologues (**Fig 5B**). Furthermore, the GisA, GisC, and Ggt homologues exhibit exceedingly low percent similarities (**Fig 5A**). This finding suggests that *S. epidermidis* is incapable of utilizing low concentrations of GSH or GSSG as sources of nutrient sulfur.

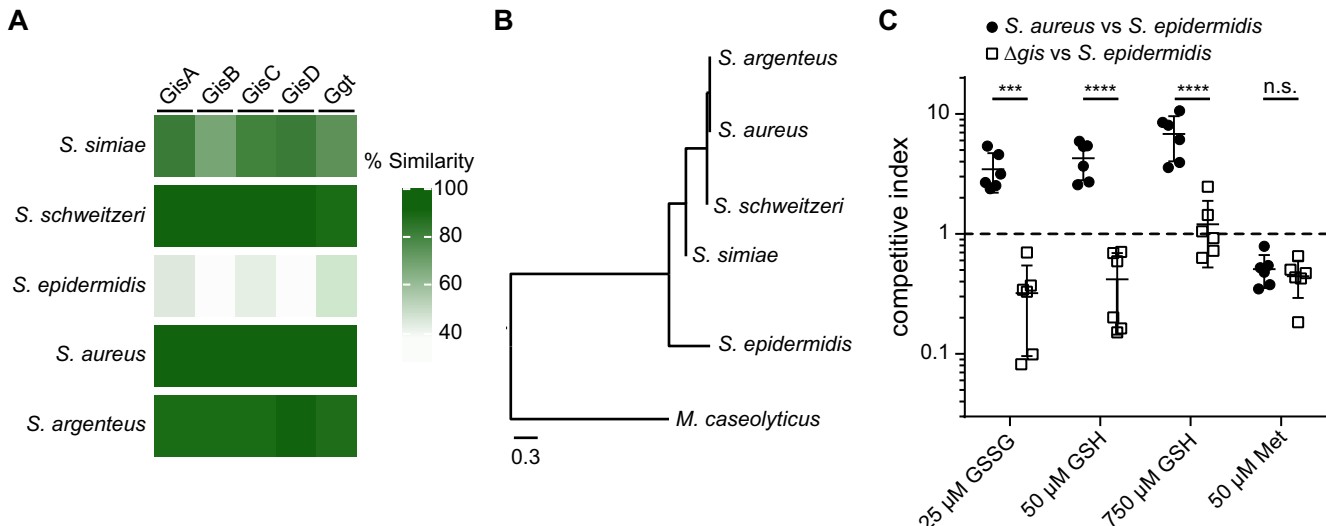

**Fig 5. GisABCD is conserved in exclusive staphylococci and promotes competition in GSH- or GSSG-supplemented media. A.** Heatmap depicting percent similarity of GisA, GisB, GisC, GisD, and Ggt proteins across select staphylococcal species (*S. aureus* USA300_FPR3757 GisA, GisB, GisC, GisD, and Ggt amino acid sequences were used as the starting point for the homology search). Conservation cross Firmicutes is depicted in **S9 Fig. B.** Phylogenic relationship between *Staphylococcus* species based on 16S rRNA sequences. **C.** *In vitro* competition experiments between *S. epidermidis* strain RP62a and WT (closed circles) or Δ*gisABCD-ggt* (Δ*gis*) *S. aureus* (open squares). The mean and standard deviation are presented. *** indicates *P*-value = 0.0003, **** indicates *P*-value <0.0001 as determined by one-way ANOVA with a Sidak multiple test correction.

## GisABCD-Ggt promotes interspecies *Staphylococcus* competition in a GSH-specific manner

To quantify *S. epidermidis* organic sulfur source-dependent proliferation, we first needed to define whether this species is capable of utilizing Met, a component of PN medium, as a source of nutrient sulfur. Additionally, *S. epidermidis* encodes predicted sulfate assimilation enzymes; thus, sulfate ($MgSO_4$) was removed from PN medium [39,40]. Supplementation of sulfate depleted PN with Met as the sole source of nutrient sulfur stimulates proliferation of *S. epidermidis*, but not *S. aureus* (**S10A** and **S10B Fig**). Therefore, Met was also removed and the resulting medium, $PN_{mod}$, was used to assess GSH- and GSSG-dependent proliferation of *S. epidermidis*. Compared to *S. aureus*, *S. epidermidis* growth exhibited considerable lag phases in media supplementation with either 25 μM GSSG or 50 μM GSH (**S10C and S10E Fig**). Increasing GSH concentrations to 750 μM resulted in comparable proliferation between *S. epidermidis* and *S. aureus* (**S10F Fig**). Increasing GSSG concentrations from 25 μM to 375 μM slightly decreased the *S. epidermidis* lag phase (**S10D Fig**). While GSH and GSSG eventually promote *S. epidermidis* growth, clearly this species struggles to proliferate when GSSG and GSH are added as sulfur sources compared to *S. aureus* unless a physiologically relevant concentration of GSH is supplied. These results are consistent with the conclusion that GisABCD-Ggt facilitates efficient *S. aureus* GSSG and GSH utilization and suggest that low affinity GSH acquisition might be conserved between the two species.

Next, the capacity of GisABCD-Ggt to provide a competitive advantage to *S. aureus* over *S. epidermidis* was determined. To evaluate this, competitive indices between *S. aureus* and *S. epidermidis* were quantified after a 24 h co-culture in $PN_{mod}$ supplemented with different sulfur sources. As expected, *S. aureus* outcompeted both a *S. epidermidis* clinical isolate and the laboratory RP62a strain in $PN_{mod}$ supplemented with 25 μM GSSG or 50 μM GSH (**Figs 5C and S10G**). Conversely, *S. epidermidis* strains outcompeted *S. aureus* in medium with 50 μM Met.

*S. aureus* exhibited a competitive advantage over *S. epidermidis* in 750 μM GSH, despite equivalent *S. epidermidis* monoculture proliferation (**Figs 5C and S10G**). Both *S. epidermidis* strains outcompeted *S. aureus* Δ*gis* in 25 μM GSSG and 50 μM GSH, underscoring the importance of GisABCD-Ggt in promoting *S. aureus* competition over *S. epidermidis* in environments containing GSH or GSSG. Equivalent quantities of *S. epidermidis* and *S. aureus* Δ*gis* were recovered in medium supplemented with 750 μM GSH (**Figs 5C and S10G**). These findings support the conclusion that Gis-independent GSH acquisition is conserved between *S. aureus* and *S. epidermidis*, while GisABCD-Ggt promotes *S. aureus* competition over *S. epidermidis*.

## Discussion

This study increases our knowledge of *S. aureus* nutrient sulfur acquisition strategies by expanding the host-derived metabolites capable of satisfying the sulfur requirement to include GSSG and identifying proteins that play a pivotal role in the use of GSSG and GSH. The proteins include an ABC transporter, which we named GisABCD, and the γ-glutamyl transpeptidase, Ggt. Lithgow *et al.* previously showed that chemically defined agar medium supplemented with GSH, CSSC, Cys, sulfide, or thiosulfate stimulated proliferation of *S. aureus* while sulfate and methionine failed to promote appreciable growth [22]. Previous investigations into the sulfur acquisition pathways used by *S. aureus* revealed strategic employment of redundancy. In fact, *S. aureus* encodes two transporters, TcyP and TcyABC, to acquire Cys and CSSC while the current study highlights multiple mechanisms of GSH acquisition and catabolism [4]. This is exemplified by the finding that Δ*gis* mutant proliferation defects in medium supplemented with 50 μM GSH can be suppressed by increasing GSH quantities to physiologically relevant concentrations. A multifaceted strategy to acquire GSH makes implicit sense given the relative abundance of GSH in host cells and because the host invests considerable energy maintaining GSH in the reduced form [12,41]. *S. aureus* likely encounters an oxidized environment as a consequence of the host innate immune response; therefore, targeting GSSG increases the number of sulfur sources available within the overall host sulfur metabolite reservoir, potentially extending the dynamic range of tissue environments and conditions conducive to staphylococcal growth. Whether the abundance GSH and GSSG change relative to each other as a result of staphylococcal infection is a focus of ongoing investigations, though a recent report showed that GSH is the fourth most abundant metabolite increased in bronchoalveolar lavage as a consequence of *S. aureus* infection [42].

While this is the first study to identify a transporter involved in *S. aureus* GSH and GSSG utilization, GSH transporters have been established in other bacteria. The first bacterial GSH transporter to be discovered was the *E. coli* ABC transporter GsiABCD (formerly *yliABCD*) [43]. Though GisABCD and GsiABCD are both members of the ABC transporter family, there are several noteworthy distinctions. First, *gsiACBD* is entirely operonic, while *gisA* is presumed to be divergently transcribed from *gisBCD-ggt*. Second, while *gisBCD* is operonic with *ggt*, GsiABCD lacks *ggt*, but encodes an ORF of unknown function called *iaaA* [43,44]. In addition, the amino acid sequences of the GisD and GsiB substrate binding lipoproteins indicate different mechanisms of GSH and GSSG recognition [45]. Together, these observations support a distinct nomenclature. Other mechanisms of GSH import have been reported in Gram-positive pathogens. For example, in *Streptococcus mutans* the substrate binding lipoprotein, GshT, promotes GSH import [46,47]. Subsequent studies revealed that GshT works in concert with TcyBC to acquire GSH and satisfy the sulfur requirement [48]. GshT homologues are encoded in other streptococci including *S. pyogenes* and *S. pneumoniae*. In *S. pneumoniae*, a Δ*gshT* mutant was shown to be more sensitive to superoxide, copper, cadmium, zinc, and innate-derived hypothiocyanous acid [13,49,50]. The Δ*gshT* mutant exhibited decreased burdens

compared to WT in the nasal cavity and blood [13]. *S. pyogenes* Δ*gshT* mutants were more sensitive to $H_2O_2$ and challenge with human neutrophils compared to WT [51]. Whether *S. aureus* scavenges GSH as a mechanism to protect against reactive oxygen species, in addition to satisfying nutritional requirements, is a focus of ongoing study.

GSH acquisition has also been explored in Gram-negative pathogens. For instance, *Francisella tularensis* is capable of replicating within macrophages and independent transposon mutagenesis screens identified Ggt as an important intracellular growth factor [3,5]. Additionally, genes harboring Tn insertions in the Cys-Gly transporter, DptA, and an alternative GSH peptidase, ChaC, also exhibited decreased intracellular proliferation within murine macrophages [5]. Together, these data support a model whereby GSH is catabolized in the *F. tularensis* periplasm by either ChaC or Ggt, generating Cys-Gly which is transported into the cytoplasm via DptA, fulfilling the sulfur requirement. The finding that Ggt and ChaC both participate in GSH catabolism reveal that multiple GSH degradation pathways can be active in a bacterial cell. Evidence of redundant mechanisms of GSH catabolism are also presented here as *S. aureus* proliferation in medium supplied with 750 uM GSH is independent of Ggt. These results suggest *S. aureus* encodes another peptidase capable of cleaving the GSH γ-peptide bond. Identification of this hypothetical peptidase is a focus of active investigation.

Ggt is conserved throughout all three kingdoms and is typically expressed extra-cytoplasmically. In humans, GGT is peripherally associated with the outer leaflet of the plasma membrane where it plays important roles in sulfur, Cys and redox homeostasis [52]. In the Gram-positive bacterial species *B. subtilis* and *B. licheniformis*, Ggt is secreted [17,19]. In Gram-negative bacteria, like *E. coli*, *Proteus mirabilis*, *and Helicobacter pylori*, Ggt is secreted to the periplasm where it partakes in GSH catabolism [16,53–55]. A notable exception is *N. meningitidis* in which Ggt resides in the cytoplasm [18]. *S. aureus* produces the low molecular weight thiol bacillithiol *in lieu* of GSH suggesting that cytoplasmic Ggt expression primarily functions in nutrient sulfur acquisition rather than GSH or Cys homeostasis [56]. However, *N. meningitidis* synthesizes GSH despite expressing Ggt within its cytoplasm [57]. These observations demonstrate that the relationship between Ggt function and localization is increasingly complex and will need to be resolved by additional analyses of Ggt localization in other bacterial species. Nonetheless, Ggt is an established virulence factor for *F. tularensis*, *N. meningitidis*, *H. pylori*, *Acinetobacter baumanii*, *Campylobacter jejuni*, and *Bacillus anthracis* [3,5,55,58–61]. Therefore, understanding whether the enzyme functions in nutrient sulfur acquisition, Cys and redox homeostasis, or both could provide a framework for designing toxic analogues that synergize with the host oxidative burst. While a role for Ggt in staphylococcal systemic infection can be ruled out, the enzyme may prove to be important in other models of pathogenesis. Alternatively, identification of the secondary GSH peptidase will help determine whether GSH catabolism is important for *S. aureus* blood stream infections.

Tracing conservation of GisABCD-Ggt across Firmicutes revealed that the system is encoded by an exclusive clade of *Staphylococcus* species which includes *S. argenteus*, *S. schweitzeri*, and *S. simiae*, but excludes *S. epidermidis* [38]. *S. epidermidis* proliferates poorly in a GSSG-supplemented medium and GisABCD-Ggt promotes *S. aureus* competition over *S. epidermidis* in GSH- and GSSG-limiting environments. These results further underscore the importance of GisABCD-Ggt to acquisition of GSSG and GSH as sources of nutrient sulfur. Both *S. aureus* and *S. epidermidis* are common residents of the human skin microflora and *S. argenteus* causes skin and soft tissue infections and sepsis in humans [62–64]. Therefore, the skin represents a dynamic host niche that could be further explored to determine whether GisABCD-Ggt provides a competitive advantage to *S. aureus* and *S. argenteus* over *S. epidermidis*. *S. epidermidis*, *S. aureus*, and *S. schweitzeri* also colonize nasal passages of humans or closely related primates [37,65]. However, the lack of GisABCD-Ggt conservation in *S.*

*epidermidis* suggests that the system is not essential for nasal colonization. Interestingly, *de novo* Met synthesis is a critical pathway for *S. aureus* nasal colonization with Cys presumably serving as the precursor [66]. Previous reports indicate negative associations between *S. aureus* and *S. epidermidis* in the nasal cavity, but accounts of co-colonized individuals have also been established [67,68]. The presence of sulfate in nasal secretions seems to favor *S. epidermidis* [66], and yet roughly 30% of the population is colonized by *S. aureus* [69]. Given that the nares represent a potential environment where nutrient sulfur competition between the species could promote niche expansion of the winner, it is tempting to speculate that the distinct nutrient sulfur sources targeted by *S. epidermidis* and *S. aureus* allow them to gain a foothold, compete, and or coexist within the complex nasal environment.

## Materials and methods

### Ethics statement

This study was conducted in meticulous accordance with the recommendations in the Guide for the Care and Use of Laboratory Animals of the National Institutes of Health. The approved protocol, PROTO201800068, was reviewed by the Animal Care and Use Committee at Michigan State University.

**Bacterial strains used in this study.** The WT *S. aureus* strain used in these studies was JE2, a laboratory derivative from the community-acquired, methicillin-resistant USA300 LAC [28]. SAUSA300_0200-*ggt* mutant strains were generated via transduction of the transposon-inactivated gene from the Nebraska Transposon Mutant Library strain into JE2 using previously described techniques [28,70]. Bacterial strains used in this study are presented in S1 Table.

A strain harboring an in-frame deletion of *gisABCD-ggt* (Δ*gis*) was constructed using a previously described allelic exchange methodology for *S. aureus* [71]. One kb upstream of SAUSA300_0200 and one kb downstream of *ggt* were amplified using primers listed in S2 Table and cloned into pKOR1-mcs. pKOR1-mcs was confirmed to have correct 1kb homology sequences by Sanger sequencing. The deletion strain was screened for hemolysis on blood agar plates and displayed WT-like hemolysis.

**Isolation of clinical isolates.** Four clinical isolates of *S. aureus* were obtained from de-identified specimens at a regional hospital clinical laboratory. Three abscess isolates were confirmed to be methicillin-resistant (strains 1055–1057) and the other was a methicillin-susceptible bone isolate (strain 1059). Identification and minimum inhibitory concentration assays were performed following Clinical and Laboratory Standards Institute approved methods. After initial isolation, subcultures were grown on tryptic soy agar (TSA, Remel) overnight.

**Sulfur source-dependent proliferation analysis.** Chemically defined (PN) medium was prepared as previously described [4,27]. PN medium was supplemented with 5 mg mL$^{-1}$ glucose for this work. Prior to inoculation in PN *S. aureus* was cultured in tryptic soy broth (TSB, BD Bacto) overnight, washed with phosphate-buffered saline (PBS), and resuspended in PN medium to an OD$_{600}$ equal to 1. Round bottom 96-well plates containing PN supplemented with 5 mg mL$^{-1}$ glucose and the indicated sulfur sources were inoculated with *S. aureus* strains at an initial inoculum of OD$_{600}$ 0.01. Growth analysis was carried out in a Biotek Epoch 2 plate reader set to 37° C with continuous shaking for the indicated time. PN was modified to test sulfur source-dependent proliferation of *S. epidermidis* and *S. aureus* by replacing MgSO$_4$ with MgCl$_2$ and omitting Met, resulting in PN$_{mod}$. *S. aureus* and *S. epidermidis* growth curves were performed as described above in PN$_{mod}$ supplemented with the indicated sulfur sources for 25 h. Sulfur sources were purchased from Millipore Sigma and GSH solutions were freshly prepared prior to each trial to limit oxidation. Alternatively, to ensure CSSC, GSH, and GSSG

were maintained in their respective reduced or oxidized forms, stock solutions were prepared by weighing the appropriate amount of the chemical aerobically and immediately transferring it to an anaerobic chamber (Coy) with a 95%:5% nitrogen:hydrogen atmosphere. Sulfur sources were then resuspended in either anaerobically acclimated water (GSH and GSSG) or anaerobically acclimated 1 N HCl (CSSC). Anaerobic proliferation was monitored within the Coy chamber using a Biotek Epoch 2 plate reader.

**Isolation of GSSG-proliferation impaired transposon mutants.** Mutant strains that comprise the Nebraska transposon mutant library (NTML) were transferred from flat bottom 96-well plates archived at -80˚C into flat bottom 96-well plates containing TSB supplemented with 10 µg mL$^{-1}$ erythromycin using a 96-pronged replica plater (Millipore Sigma). Mutant cells were cultured overnight at 37˚C and subcultured using a 1:150 dilution into a fresh round bottom 96-well plate containing 150 µL PN media supplemented with 25 µM GSSG instead of 25 µM CSSC as the primary sulfur source. Proliferation of each mutant within the 96-well plate was assessed by monitoring OD$_{600}$ over 22 h using a H1 Biotek plate reader set to 37˚C with continuous shaking. This process was repeated for all twenty 96-well plates that comprise the NTML. To account for plate-to-plate variation, proliferation was averaged across all 96 transposon (Tn) mutant strains within a given 96-well plate. Tn mutants that exhibited decreased proliferation equal to at least two standard deviations below the 96-well plate average across mid-exponential and stationary phase were isolated by streak plating onto TSA supplemented with 10 µg mL$^{-1}$ erythromycin. Isolated Tn mutants were validated in a secondary screen that compared proliferation in PN media supplemented with 25 µM GSSG to WT JE2 in technical and biological triplicate using round bottom 96-well plates and monitoring OD$_{600}$ over 22 h using a H1 Biotek plate reader set to 37˚C with continuous shaking. Growth in TSB was used as a positive proliferation control. The following Tn mutants displayed significantly reduced growth compared to WT JE2 in 25 µM GSSG supplemented PN but not TSB: NE392 (*SAUSA300_0200*::Tn), NE541 (*SAUSA300_0201*::Tn), NE457 (*SAUSA300_0202*::Tn), NE215 (*SAUSA300_0203*::Tn), NE254 (*ggt*::Tn) (**S1 Table**). These mutants were selected for two additional steps of identification and validation. First, the location of the *bursa aurealis* Tn insertion for each mutant strain was verified using a previously described inverse PCR method followed by Sanger sequencing [28]. Second, the Tn mutations were backcrossed into the WT JE2 parental strain using a previously described phage transduction method [28,70] and the resulting backcrossed mutants were analyzed for proliferation as mentioned above in PN supplemented with 25 µM GSSG or the indicated sulfur source (**S1 Table**).

**Liquid Chromatography-Tandem Mass spectrometry quantification of *S. aureus*-associated GSSG.** WT JE2, Δ*gis* and *ggt*::Tn strains were cultured in a 1 L Erlenmeyer flask containing 250 mL PN supplemented with 25 µM CSSC (4:1 flask-to-volume ratio). Cultures were incubated at 37˚C with 225 rpm to an OD$_{600}$ of 0.4–0.5. At this time the three cultures (WT JE2, Δ*gis* and *ggt*::Tn) were split into two 50 mL conical tubes each containing 50 mL aliquots of culture, the cells washed once in PBS, and resuspended in 50 mL of prewarmed PN media supplemented with either 0 µM or 25 µM GSSG. The six 50 mL conical tubes were incubated at 37˚C without shaking for 5 min. Cells were centrifuged, washed twice with 10 mL PBS, and resuspended in 500 µL HPLC grade methanol containing [$^2$H$_{10}$] GSSG (Toronto Research Chemicals, Canada) as the internal standard. Cells were lysed via bead beating using 0.1 µm zirconia beads (BioSpec, Bartlesville, OK) in a Bullet Blender Storm 24 (Next Advanced, Troy, NY) at maximum speed for 2 min. The lysate was collected and centrifuged at maximum speed for 10 min to remove insoluble debris. Metabolite-containing supernatants for each of the six samples were dried using a Savant DNA 120 SpeedVac concentrator (Thermo Scientific) and were stored overnight at -20˚C. Samples were reconstituted 500 µL of 0.1% formic acid in preparation for mass spectrometry.

Metabolites from a 10 μL aliquot from each sample were separated in reversed-phase mode using an Acquity HSS T3 column (1.8 μm 100 X 2.1 mm$^2$; Waters, Milfor, MA). Solvent A was 0.1% formic acid in water and solvent B was methanol. The mobile phase gradient was as follows: 0 min−100% A, 1 min − 100% A, 5 min − 60% A, 7 min − 1% A, 8 min − 1% A; 8.01 min − 100% A, and 10 min − 100% A with a flow rate of 0.3 mL/min. Mass spectrometry detection was performed using a Xevo G2-XS quadrupole time-of-flight (QTof) by positive ion electrospray ionization. GSSG detection was based on the retention time of 2.96 min to 3.00 min and mass accuracy using MassLynx Version 4.2 (Waters). GSSG calibration curves were generated with a six-point curve of serially diluted unlabeled GSSG standards with the corresponding concentration of [$^2$H$_{10}$] GSSG.

**Expression and purification of Ggt and GisA.** Open reading frames corresponding to the *ggt* or *gisA* genes were PCR amplified from the *S. aureus* JE2 genome with primers listed in **S2 Table**. Subsequently, the genes were cloned into the pET28b expression vector using NEB HiFi Gibson assembly kit (NEB, New England, MA) after the plasmid has been linearized with NcoI-HF and XhoI-HF. The assembly mixture was transformed into *E. coli*, cells were recovered in lysogeny broth (LB), and plated onto LB agar (Fisher) containing 50 μg mL$^{-1}$ kanamycin and 5 mg mL$^{-1}$ glucose. Plasmids were confirmed using Sanger sequencing and transformed into an *E. coli* NEB strain 3016 *slyD* mutant [72]. Transformed *E. coli* were cultured in LB with 50 μg mL$^{-1}$ kanamycin overnight at 37˚ C with shaking at 225 rpm, sub-cultured 1:50 into 500 mL LB with 50 μg mL$^{-1}$ kanamycin in a 2 L flask and grown to an OD$_{600}$ of 0.4–0.7. Ggt or GisA protein expression was induced by addition of 200 μM isopropyl-1-thio-β-D-galactopyranoside (IPTG) and the culture was separated into five 500 mL flasks containing 100 mL of culture and incubated for 4 h at 27˚ C and 225 rpm shaking. After induction, cells were centrifuged at 10,000 x g for 10 min at 4˚ C and washed with PBS. Resulting GisA and Ggt induction pellets were resuspended in 40 mL of buffer containing 50 mM tris, 200 mM KCl, 20 mM imidazole at pH 8, or 40-mL buffer containing 50 mM tris, 500 mM NaCl, 20 mM imidazole at pH 8, respectively. Cells were lysed via five consecutive cycles through a fluidizer set to 20,000 psi. Lysates were then centrifuged at 15,000 x g for 15 min to remove intact cells and the resulting supernatant was retained. To purify the target proteins, Ni-NTA chromatography was used. Purification was performed by incubating the cleared lysate with 1 mL Ni-NTA resin (Qiagen, Hilden, Germany) on a rotating platform at 4˚ C for 2 h. Protein was eluted with 50 mM tris 400 mM imidazole. Buffers used to purify GisA contained 200 mM KCl while buffers used to purify Ggt contained 500 mM NaCl. Each buffer contained 1x protease inhibitor cocktail (Millipore-Sigma). The GisA elutant was dialyzed using 10 mM tris, 200 mM KCl at pH 7.5 as the dialysis buffer for 18 h. The Ggt elutant was dialyzed using 10 mM tris, 150 mM at pH 7.0 as the dialysis buffer for 18 h. Both elutions were concentrated using a 10 kDa molecular weight cutoff protein concentrators. Purification was confirmed via electrophoresis using 12% SDS-PAGE gels. Protein concentrations were determined with the bicinchoninic acid (BCA) protein kit (Pierce ThermoFisher).

**Quantitation of Ggt enzyme kinetics.** γ-glutamyl transpeptidase reactions contained 5 μg recombinant Ggt, reaction buffer (10 mM tris with 150 mM NaCl), and the indicated concentrations of GSH and GSSG dissolved in reaction buffer. Reactions proceeded for 30 min at 37˚ C after which samples were incubated at 80˚ C for 5 min to stop the reaction. Samples were dried using a roto-vac speed vacuum and stored at -80˚ C until they were hydrated via resuspension in water, derivatized with carboxybenzyl (CBZ), and applied to a Waters Xevo TQ-S triple quadrupole mass spectrometer as previously described [73]. Peak processing was performed by MAVEN, and the signal was normalized to a $^{13}$C-glutamine internal standard [74]. An external glutamate standard curve was generated using the same chromatographic conditions, and the signal was normalized to a $^{13}$C-glutamine internal standard. A fit equation

to the standard curve was employed to quantify glutamate within the samples. Glutamate released per min was calculated and data were fit to the Michaelis-Menten equation using GraphPad Prism. Data represent the average of glutamate quantified from four independent protein purifications.

**Fractionation and western blot analysis of His-tagged *S. aureus* Ggt.** The His-tagged *ggt* ORF was amplified from pET28b::*ggt* and cloned into pOS1 P$_{lgt}$ digested with NdeI and HindIII using Gibson assembly to generate Ggt-His (His +). The *ggt* ORF was amplified from JE2 genomic DNA and pOS1 P$_{lgt}$ digested with NdeI and HindIII using Gibson assembly to generate the untagged Ggt (His -). Plasmids were confirmed by Sanger sequencing and transformed from *E. coli* DH5α into *S. aureus* RN4220 via electroporation. Plasmids were purified from RN4220 and transformed into JE2 *ggt*::Tn. An empty vector control strain was generated by transforming JE2 and *ggt*::Tn with pOS1 P$_{lgt}$. To assess Ggt localization, *S. aureus ggt*::Tn pOS1 P$_{lgt}$::*ggt* (His -) and *ggt*::Tn pOS1 P$_{lgt}$::*ggt*-His (His +) cultures were prepared as previously described in the proliferation analysis section by sub-culturing into three, 250 mL flasks each containing 100 mL PN supplemented 25 μM GSSG and 10 μg mL$^{-1}$ chloramphenicol at a starting OD$_{600}$ equal to 0.1. Cells were cultured for 4 h at 37˚ C and 225 rpm shaking to mid-exponential phase. At this time cells were collected via centrifugation, the supernatant recovered, and the cell pellet washed with PBS. A total of 50 mL of supernatant was precipitated with trichloracetic acid (TCA, final percent of 10% v/v), incubated overnight at 4˚ C, pelleted, and the resulting pellet washed twice with 95% ethanol. Whole cell lysates (WCL) were prepared by resuspending the washed cell pellet in membrane buffer (50 mM Tris-HCl pH 7.0, 10 mM MgCl$_2$, 60 mM KCl) and transferring the suspension to a 2 mL bead beating tube containing 500 μL volume of 0.1 mm zirconia/silica beads (BioSpec Bartlesville, OK). Cells were bead beaten using a Mini-beadbeater 16 (BioSpec) thrice for 1 min and centrifuged 4,000 x g for 10 min to remove cells that were not lysed. Cell wall and lysed protoplast fractions were generated from whole cells cultured as described above. Staphylococci were pelleted via centrifugation, resuspended in TSM (100 mM Tris-HCl pH 7.0, 500 mM sucrose, 10 mM MgCl$_2$), and incubated with 100 μg lysostaphin for 1 h at 37˚ C. Protoplasts were recovered by centrifugation at 13,000 x g for 15 min. The supernatant containing the cell wall fraction was collected and the protoplasts were washed twice with TSM prior to resuspension in membrane buffer and lysed via bead beating as previously described. Intact protoplasts were removed via centrifugation at 4,000 x g for 10 min, generating the protoplast lysate. Protein concentrations of the whole cell lysate (WCL), cell wall, and protoplast lysate were determined using the BCA method (Pierce ThermoFisher). The cell wall fraction was further concentrated using the TCA precipitation method previously described for the supernatant. WCL and lysed protoplast fractions were mixed 1:1 by volume with 2x Laemmli buffer, boiled for 10 min, and an equivalent of 40 μg was loaded onto a 12% SDS-PAGE gel. Concentrated cell wall pellet was resuspended in 1x Laemmli buffer, boiled for 10 min, and 40 μg was loaded onto a 12% SDS-PAGE gel. PAGE was performed using Tris-glycine running buffer and samples were transferred at 65 volts for 1 h to a PVDF membrane (GVS North America) at 4˚ C. Membranes were incubated overnight in phosphate buffered saline tween-20 (PBST) with 3% bovine serum albumin (BSA) at 4˚ C with agitation. An αHis mouse antibody (Millipore) was used as the primary antibody at a 1:4,000 dilution in PBST supplemented with 5% BSA and incubated for 1 h with shaking. The membrane was washed thrice with PBST. An α-mouse IgG conjugated to horseradish peroxidase (HRP) was used as the secondary antibody at a dilution of 1:4,000 (Sigma-Aldrich). To assess supernatant and WCL fraction a rabbit anti-α-hemolysin (αHla) primary antibody (Sigma-Aldrich) was used at a 1:8,000 dilution. To control for proper cell wall fractionation a mouse anti-protein A (αSpa) primary antibody (Sigma-Aldrich) was used at 1:6,000 dilution. Membranes were washed thrice in PBST after incubation with either a 1:10,000 diluted

horseradish peroxidase (HRP)- linked goat anti-rabbit IgG (Sigma) or 1:5,000 diluted HRP-linked goat anti-mouse IgG (Millipore) secondary antibodies. Membranes were developed using the ECL Prime kit (Cytiva, Marlborough, MA) and imaged using Amersham Imager 600 (GE Healthcare, Amersham, Buckinghamshire, UK). Fractionation was repeated with three independent sets of cultures and one set of immunoblots is presented.

**Proteinase K protection of *S. aureus* Ggt.**   *S. aureus ggt*::Tn pOS1 P$_{lgt}$::*ggt* and *ggt*::Tn pOS1 P$_{lgt}$::*ggt*-His were cultured as described in the western blot analysis section. Cells were collected via centrifugation after normalizing OD$_{600}$. The resulting pellet was washed with PBS, resuspended in TSM, and incubated with 100 μg lysostaphin for 1 h at 37˚C prior to treatment with 10 μg mL$^{-1}$ proteinase K (Thermo-Scientific) for 30 min at 37˚C. At this time 5 mM phenylmethylsulfonyl fluoride was added to halt Proteinase K activity. Protoplasts were isolated from the cell wall fraction via centrifugation at 13,000 x g at 4˚C for 15 min and washed twice with TSM. The cell wall fraction protein concentration was determined using BCA and was TCA precipitated as previously described. Protoplasts were lysed, and protein concentrated using the Wizard Genomic DNA Purification Kit (Promega) nuclei lysis buffer and protein precipitation solution, respectively. Cell wall and protoplast protein pellets were reconstituted in Laemmli buffer and immunoblotted as described above.

**Identification of GisABCD-Ggt homologues across bacteria.**   The USA300_FPR3757 (assembly GCF_000013465.1) Ggt protein sequence (ABD22038.1) was used as the query protein for homology searches with MolEvolvR using DELTA-BLAST and the NCBI RefSeq database [36,75–77]. Data were filtered to include only Firmicutes that encoded Ggt homologues containing a glutamyl transpeptidase domain. Only genomes harboring Ggt homologues were used to query USA300_FPR3757 GisABCD. Percent similarities to the *S. aureus* GisABCD-Ggt protein amino acid sequences were used to generate a heatmap. The heatmap and hierarchical clustering of similar protein profiles were generated using the R package, pheatmap.

**16S rRNA phylogenic analysis of Staphylococcal species.**   16S ribosomal RNA DNA sequences were retrieved from NCBI for the following strains: *S. aureus* USA300 FPR3757, *S. epidermidis* RP62s, *S. simiae* NCTC 13838, *S. schweitzeri* NCTC13712, and *S. argenteus* 58113. Parsimonious reconstruction was conducted using kSNP4 using default parameters and selecting *Macrococcus caseolyticus* FDAARGOS_868 as the outgroup [78,79].

**Quantitation of *S. aureus* and *S. epidermidis* competition.**   *S. aureus*, *S. aureus Δgis*, and *S. epidermidis* were cultured overnight in TSB, pelleted, washed in PBS, and normalized to the same OD$_{600}$ in PN$_{mod}$. Strains were mixed in a 1:1 ratio (v/v) and inoculated into 5 mL PN$_{mod}$. PN$_{mod}$ was supplemented with 25 μM GSSG, 50 μM GSH, 750 μM GSH, or 50 μM Met. Dilution plating of the freshly mixed co-culture was plated onto mannitol salt agar (MSA) to quantify initial counts of each organism. Cultures were incubated for 24 h at 37˚C with 225 rpm shaking after which the cultures were dilution plated onto MSA and allowed to grow for 48 h at 35˚C. *S. aureus* ferments mannitol and appears yellow on MSA, while *S. epidermidis* does not and maintains a pink color; consequently, yellow and pink colored colonies were enumerated to assess quantities of each organism. Competitive indices (CI) were calculated by dividing the *S. aureus* to *S. epidermidis* output ratio by the *S. aureus* to *S. epidermidis* input ratio. A CI greater than one indicates more *S. aureus* than *S. epidermidis* while a CI less than one signifies greater quantities of *S. epidermidis* compared to *S. aureus*.

## Supporting information

**S1 Text. Supporting Materials and Methods.**
(DOCX)

**S1 Table. *Staphylococcus* strains used in this study.**
(DOCX)

**S2 Table. Primers used in this study.**
(DOCX)

**S1 Fig. GSSG supplementation as the sole source of nutrient sulfur stimulates proliferation of *S. aureus*.**
(DOCX)

**S2 Fig. GisABCD-Ggt promotes anaerobic proliferation in PN medium supplemented with GSSG or GSH.**
(DOCX)

**S3 Fig. Ectopic expression of native or His-tagged Ggt complements *ggt* mutant proliferation in medium supplemented with reduced or oxidized GSH.**
(DOCX)

**S4 Fig. Domain architectures and secondary structure predictions for the *S. aureus* GisABCD-Ggt system.**
(DOCX)

**S5 Fig. GisA encodes ATPase domain signatures and demonstrates ATP hydrolysis activity.**
(DOCX)

**S6 Fig. Heterologous expression and purification of *S. aureus* Ggt from *Escherichia coli*.**
(DOCX)

**S7 Fig. Virulence of a *gisB*::Tn mutant strain mimics wild type.**
(DOCX)

**S8 Fig. *S. aureus* acquires GSH independent of GisABCD-Ggt in physiologically relevant concentrations of GSH.**
(DOCX)

**S9 Fig. Conservation of Ggt and GisABCD across Firmicutes.**
(DOCX)

**S10 Fig. *S. epidermidis* and *S. aureus* nutrient sulfur source utilization is distinct and promotes interspecies competition.**
(DOCX)

**S1 Data Table. Raw data that support the graphs in the figures.**
(XLSX)

## Acknowledgments

The defined transposon mutant library used in this study was provided by the Network on Antimicrobial Resistance in *Staphylococcus aureus* (NARSA) for distribution by BEI Resources, NIAID, NIH: Nebraska Transposon Mutant Library (NTML) Screening Array NR-48501. We thank the laboratory of Dr. Taeok Bae at Indiana University for supplying the pKOR1-mcs plasmid, Dr. Anthony Richardson for the *S. epidermidis* strain RP62a, and we thank the Dr. Victor DiRita and Dr. Sean Crosson laboratories at Michigan State University for technical support. We also thank Dr. Martin P. Ogrodzinski, Dr. Robert Quinn, Doug

Guzior, Dr. Anthony Schilmiller, and the MSU Mass Spectrometry and Metabolomics Core for technical support.

## Author Contributions

**Conceptualization:** Michael R. Wischer, Cristina Kraemer-Zimpel, Janani Ravi, Neal D. Hammer.

**Data curation:** Joshua M. Lensmire, Michael R. Wischer, Cristina Kraemer-Zimpel, Paige J. Kies, Lo Sosinski, Elliot Ensink, Daniel H. Havlichek, Jr., Janani Ravi, Neal D. Hammer.

**Formal analysis:** Joshua M. Lensmire, Cristina Kraemer-Zimpel, Paige J. Kies, Neal D. Hammer.

**Funding acquisition:** Sophia Y. Lunt, Neal D. Hammer.

**Investigation:** Joshua M. Lensmire, Michael R. Wischer, Cristina Kraemer-Zimpel, Paige J. Kies, Lo Sosinski, Elliot Ensink, Jack P. Dodson, John C. Shook, Phillip C. Delekta, Christopher C. Cooper, Martha H. Mulks, Neal D. Hammer.

**Methodology:** Joshua M. Lensmire, Michael R. Wischer, Cristina Kraemer-Zimpel, Paige J. Kies, Lo Sosinski, Elliot Ensink, John C. Shook, Christopher C. Cooper, Daniel H. Havlichek, Jr., Martha H. Mulks, Sophia Y. Lunt, Janani Ravi, Neal D. Hammer.

**Resources:** Sophia Y. Lunt, Neal D. Hammer.

**Software:** Janani Ravi.

**Supervision:** Neal D. Hammer.

**Writing – original draft:** Joshua M. Lensmire, Neal D. Hammer.

**Writing – review & editing:** Joshua M. Lensmire, Michael R. Wischer, Cristina Kraemer-Zimpel, Paige J. Kies, Phillip C. Delekta, Christopher C. Cooper, Daniel H. Havlichek, Jr., Martha H. Mulks, Sophia Y. Lunt, Janani Ravi, Neal D. Hammer.

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
