## [Decision Letter · Decision Letter 0]

4 Apr 2023

Dear Dr Hammer,

Thank you very much for submitting your Research Article entitled 'The glutathione import system satisfies the Staphylococcus aureus nutrient sulfur requirement and promotes interspecies competition.' to PLOS Genetics.

The manuscript was fully evaluated at the editorial level and by independent peer reviewers. The reviewers appreciated the attention to an important topic but identified some concerns that we ask you address in a revised manuscript.

We therefore ask you to modify the manuscript according to the review recommendations. Your revisions should address the specific points made by each reviewer.

Yours sincerely,

Aimee Shen

Academic Editor

PLOS Genetics

Lotte Søgaard-Andersen

Section Editor

PLOS Genetics

The Reviewers were unanimous in their view that this manuscript is an important contribution to our understanding of how Staphylococcus aureus acquires sulfur from its environment by metabolizing glutathione imported by a glutathione transport system identified by the authors. However, they have suggested a few experiments for the Authors to consider that would provide additional rigor to your carefully conducted study. Reviewers 1 and 3 indicated that demonstrating direct transport of GSH, e.g. using a 3H-GSH-GSSG, would allow the Authors to more forcefully conclude that they have identified the GSH transporter. Nevertheless, his experiment is optional given that it might be challenging to develop such an assay. Exploring the specificity of the transport system by assessing its role in taking up trip-peptides could also advance our understanding of the likely GSH transport system you have identified. Beyond these optional experiments, please address the comments and questions of the Reviewers in a revised manuscript and provide a detailed response to each Reviewer's comments in a Response to Review, which includes moving the model figure (Fig S7) to the main manuscript.

A few minor points:

Line 52: please add a comma after "operon"

Line 94: please add "it" after "but"

Line 296: please delete "compared to" at the end of the sentence

Please also add a little more detail about how the transposon screening was done to the Materials and Methods.

Reviewer's Responses to Questions

**Comments to the Authors:**

Reviewer #1: PGENETICS-D-23-00230

In this manuscript, Lensmire et al. assess a mechanism that promotes glutathione-dependent growth of Staphylococcus aureus. S. aureus uses a putative ABC-type transporter (Gis) to import reduced or oxidized glutathione. Ggt, which is a GSH gamma-glutamyl transpeptidase encoded downstream of the transporter genes, then releases glutamate within the cytoplasm so the residual peptide can facilitate cysteine biosynthesis. The Gis and Ggt factors do not affect pathogenesis but enables S. aureus to grow in GSH-rich environments and competitively exclude non-gis encoding taxa like S. epidermidis.

Overall, the manuscript is well written and will be of great interest to the study of pathogens, in which sulfur utilization is under-investigated. A few areas that may improve the manuscript:

1) Please demonstrate GSH/GSSG transport by the Gis system. Otherwise, renaming the genes and ascribing function is premature, and the predicted factors should be labeled as ‘putative’.

2) Please expand on the Gis/Ggt link that is suggested by Fig. S7. Are there thoughts on potential protein-protein interactions between Ggt and the transporter, with imported GSH/GSSG immediately cleaved? Along these lines, please include in the main text this or a similar cartoon of the importer and Ggt, with proposed mechanisms that fuel sulfur-based growth physiology.

3) Please reconsider the competition data because it seems relatively contrived, with the outcomes dictated by how species individually grow, not their interactions. While I don’t mean to propose a gain-of-function experiment, are the authors suggesting that acquisition of gis genes by S. epidermidis would enable the species to grow on GSH?

4) In Fig. 4, please incorporate a simple 16S tree to organize the species (along with an appropriate outgroup).

5) Lines 80-1: This statement seems inherently incorrect. There are many non-pathogenic microbes for which sulfur acquisition mechanisms have been explored, e.g. E. coli, various symbiotic microbes, and certainly marine organisms.

6) To what extent does overexpression of Ggt affect growth of the gis mutants in GSH?

7) For the species that lack Gis, can gene synteny reveal whether gis undergoes horizontal gene transfer? Do the genes surrounding gis/ggt in S. aureus neighbor each other in S. epidermidis?

Reviewer #2: This is a well written manuscript documenting that GisABCD functions as a GSH-GSSG transporter. Further, enzymatic assays determined that S. aureus encodes a gamma glutamyl transpeptidase (Ggt) that cleaves glutamate from GSH or GSSG facilitating utilization of cysteine. S. aureus does not synthesize glutathione but instead synthesizes a separate low molecular weight thiol called bacillithiol. Therefore, it is hypothesized that S. aureus evolved to utilize the GisABCD transporter plus Ggt to acquire cysteine from GSH-GSSG, which is abundant in a mammalian host. Most glutamyl transpeptidases are secreted, however, it was rigorously determined that Ggt in S. aureus is cytoplasmic. Further, it was determined that GisABCD/Ggt mutant does not have a bacterial burden defect in a mouse model of infection, which is surprising, and suggests that the cysteine transporters TcyP and TcyABC may complement this mutation. Lastly, it was determined that S. epidermidis and S. aureus transport disparate sulfur sources but it is clear S. epidermidis does not have the ability to utilize GSH-GSSG as a cysteine source. This manuscript is impactful and timely as we know little about how S. aureus acquires sulfur nor do we fully understand cysteine catabolism/biosynthesis. I just have a few comments.

1. I am intrigued by the growth curve in PN medium lacking sulfur in Figure 1A. Growth in this medium has a similar growth rate as PN medium containing GSH-GSSG for a short period of time but then abruptly levels off and does not have further exponential growth. What is the source of the sulfur facilitating early exponential growth? Is it contamination of sulfur in the water or potentially methionine?

2. Although not necessarily essential for this manuscript, it would add significant rigor if a transport assay would be performed with 3H GSH-GSSG to document that GisABCD is a transporter. Further, is it possible to construct a gisABCD-ggt/tcyP/tcyABC mutant? Or is that mutant not viable? This would be interesting and a good fit for this manuscript regarding the animal model of infection.

3. Methicillin-susceptible throughout instead of methicillin-sensitive

4. Streptococcus pneumoniae throughout instead of Streptococcus pneumonia

Reviewer #3: The manuscript entitled “The glutathione import system satisfies the Staphylococcus aureus nutrient sulfur requirement and promotes interspecies competition” by Lensmire, JM et al. describes a locus that encodes an ABC transporter for GSH and GSSG as well as the first enzyme in the breakdown of GSH, Ggt. The authors demonstrate substrate specificity and calculate Michaelis-Menten parameters for each substrate. The authors also demonstrate the limited distribution of similar systems among a select clade of Firmicutes. While the Gis transport system does not seem to directly contribute to overall virulence in S. aureus, it does seem to promote competition between species that lack such transporters. This, however, is difficult to put into real world context as S. aureus would most likely encounter species that lack Gis while colonizing the skin surface, an environment not known to be replete with GSH. Finally, the authors localize Ggt to the cytosol, a rare compartmentalization for such enzymes that are usually found extracellularly or in the periplasm. A few minor additions might improve the impact of the manuscript:

1. Do the authors know anything about the enzyme that separates Gly from Cys after the reaction with Ggt or can S. aureus use the sulfur directly from the Cys-Gly dipeptide?

2. How is the Gis system regulated transcriptionally? Is anything known about what induces the system?

3. Instead of screening for other sulfur sources that could be transported by Gis, are other tri-peptides transported by the system? It may not be selective for solely sulfur containing peptides.

4. With the inframe deletion mutant in the Gis locus, a screen of the Tn library should be able to easily identify the second low affinity GSH transporter. This would add dramatically to the impact of the story.

**Have all data underlying the figures and results presented in the manuscript been provided?**

Reviewer #1: **No: **I did not see any spreadsheets with the data provided in the manuscript

Reviewer #2: Yes

Reviewer #3: Yes

PLOS authors have the option to publish the peer review history of their article (what does this mean?). If published, this will include your full peer review and any attached files.

Reviewer #1: No

Reviewer #2: No

Reviewer #3: No

---

## [Editor Report · Decision Letter 1]

21 Jun 2023

Dear Dr Hammer,

We are pleased to inform you that your manuscript entitled "The glutathione import system satisfies the Staphylococcus aureus nutrient sulfur requirement and promotes interspecies competition." has been editorially accepted for publication in PLOS Genetics. Congratulations!

Yours sincerely,

Aimee Shen

Academic Editor

PLOS Genetics

Lotte Søgaard-Andersen

Section Editor

PLOS Genetics

Comments from the reviewers (if applicable):

Thank you for addressing so many of the Reviewers' comments. This is a really beautiful study.

**Data Deposition**

http://datadryad.org/submit?journalID=pgenetics&manu=PGENETICS-D-23-00230R1

**Press Queries**

---

## [Editor Report · Acceptance letter]

30 Jun 2023

PGENETICS-D-23-00230R1 

The glutathione import system satisfies the Staphylococcus aureus nutrient sulfur requirement and promotes interspecies competition. 

Dear Dr Hammer, 

We are pleased to inform you that your manuscript entitled "The glutathione import system satisfies the Staphylococcus aureus nutrient sulfur requirement and promotes interspecies competition." has been formally accepted for publication in PLOS Genetics! Your manuscript is now with our production department and you will be notified of the publication date in due course.

With kind regards,

Zsofi Zombor

PLOS Genetics

On behalf of:
